# Multi-Object Tracking on SWIR Images for City Surveillance in an Edge-Computing Environment

**DOI:** 10.3390/s23146373

**Published:** 2023-07-13

**Authors:** Jihun Park, Jinseok Hong, Wooil Shim, Dae-Jin Jung

**Affiliations:** A2Mind Inc., Daejeon 34087, Republic of Korea; jhpark@a2mind.com (J.P.); jshong@a2mind.com (J.H.); wishim@a2mind.com (W.S.)

**Keywords:** deep learning, object detection, short-wave infrared, multi-object tracking, edge computing, neural network quantization

## Abstract

Although Short-Wave Infrared (SWIR) sensors have advantages in terms of robustness in bad weather and low-light conditions, the SWIR images have not been well studied for automated object detection and tracking systems. The majority of previous multi-object tracking studies have focused on pedestrian tracking in visible-spectrum images, but tracking different types of vehicles is also important in city-surveillance scenarios. In addition, the previous studies were based on high-computing-power environments such as GPU workstations or servers, but edge computing should be considered to reduce network bandwidth usage and privacy concerns in city-surveillance scenarios. In this paper, we propose a fast and effective multi-object tracking method, called Multi-Class Distance-based Tracking (MCDTrack), on SWIR images of city-surveillance scenarios in a low-power and low-computation edge-computing environment. Eight-bit integer quantized object detection models are used, and simple distance and IoU-based similarity scores are employed to realize effective multi-object tracking in an edge-computing environment. Our MCDTrack is not only superior to previous multi-object tracking methods but also shows high tracking accuracy of 77.5% MOTA and 80.2% IDF1 although the object detection and tracking are performed on the edge-computing device. Our study results indicate that a robust city-surveillance solution can be developed based on the edge-computing environment and low-frame-rate SWIR images.

## 1. Introduction

Deep neural networks have significantly improved object detection and tracking systems. Advanced object detection and tracking algorithms have been utilized in several automatic recognition systems, including surveillance systems for military and urban areas. Although most previous studies have relied on publicly available visible image datasets, more recent studies have highlighted the potential of using Short-Wave Infrared (SWIR) sensors for automated target detection and tracking. SWIR sensors, for instance, have been used to identify UAVs [1,2], detect gas leaks [3], and conduct long-range surveillance [4]. Due to their greater sensitivity to longer light wavelengths than visible sensors, SWIR sensors can see through dense fog and thin layers. The characteristics of SWIR sensors provide advantages in marine haze penetration, atmospheric transmission, and target discrimination against backgrounds [5]. These benefits can be greatly exploited in urban surveillance scenarios that suffer from haze interference, long-range surveillance difficulties, and low-contrast image conditions. Even though SWIR sensors have such advantages, SWIR images have rarely been investigated for object detection and tracking algorithms. In this paper, we utilize SWIR sensors for city surveillance to take advantage of their robustness in bad weather and low-light conditions.

Previous multi-object tracking studies focused mostly on tracking pedestrians on sidewalks. These approaches were useful for tracking people near a car, but they are insufficient for tracking numerous classes of objects in city-surveillance environments. In a city-surveillance scenario, ground vehicle classes such as cars, trucks, motorcycles, and buses should also be detected, and they should be tracked in subsequent image frames.

As computer vision technology develops, a method of applying this technology to a low-power, low-computation computer near a sensor has been introduced, which is called edge computing. When using the edge-computing scheme, there is no need to transmit all images to the server, which reduces network load and reduces the risk of hacking issues due to network intrusion or invasion of privacy through the spread of images. In this study, we conducted experiments with Raspberry Pi 4B, a representative edge-computing device, to study object detection and tracking in SWIR images based on edge-computing.

Previous multi-object tracking approaches [6,7] have used a deep neural network-based object detector and the bounding box association method to track the object boxes of consecutive time frames. Object boxes were associated using a score function based on intersection over union (IoU) or a similarity score based on feature embedding from neural networks.

In a city-surveillance scenario, ground vehicles are much faster than people or animals, which have been the main target of multi-object tracking algorithms. In addition, since the speed of object detection is slow in the edge-computing environment, tracking in a low-frame-rate environment is required. When the frame rate is low, high-speed objects in consecutive time frames are less likely to overlap, which degrades the performance of the existing IoU-based tracking algorithms. Furthermore, the relative sizes and moving speeds vary for each class, so a predefined setting of the Kalman filter would not be adequate to track all classes in the city-surveillance scenario.

In this paper, we propose MCDTrack (Multi-class Distance-based Tracking) for the multi-object detection and tracking problem on SWIR images with an edge-computing device in a city-surveillance scenario. To realize a practical city-surveillance application based on SWIR sensors and the edge-computing environment, we adopt efficient but effective object detection methods, and we also propose a novel tracking algorithm that is suitable for our use of low-frame-rate SWIR images and low computing power. Recent lightweight object detectors that are accelerated by 8-bit integer quantization are compared for use in our multi-object tracker. MCDTrack associates object boxes in two rounds. The first round uses high-confidence boxes with a distance-based similarity score based on the distance thresholds of each class, and the second round uses the IoU similarity score. The effects of experimental parameters, such as the choice of object detector and the confidence thresholds, are investigated. Our MCDTrack achieves 77.5% in MOTA, 80.2% in IDF1, 77.2% in mMOTA, and 81.6% in mIDF1, which will also be used in real city-surveillance scenarios.

## 2. Related Work

This section introduces related work on utilizing SWIR sensors for automated object detection and tracking, and the research trends on object detection and multi-object tracking.

### 2.1. Short-Wave Infrared Sensors

There have been several attempts to utilize SWIR sensors in automated analysis. Rankin and Matthies [8] proposed a method of detecting mud areas by attaching multiple sensors, including a SWIR sensor, to a military UGV and analyzing dry and wet soil using the characteristics of the SWIR sensor in an off-road environment. Lemoff et al. [9] proposed a SWIR imaging system that uses facial recognition to generate facial images at distances up to 350m and can detect a person at a distance of hundreds of meters. Kandylakis et al. [10] developed a multi-sensor camera system consisting of SWIR, thermal, and hyperspectral cameras and detected moving objects using object detection algorithms, including Faster R-CNN [11] and YOLOv2 [12]. Pavlović et al. [4] proposed a method for automatic cross-spectral annotation of SWIR sensor images and a deep learning model for detecting two types of objects: cars and people. For SWIR sensor data annotation, Pavlović et al. [4] recorded visible-light sensor and SWIR sensor data in parallel, and performed automatic cross-spectral annotation using the YOLOX model. As an extension of these studies, our work investigates object detection in city-surveillance scenarios using SWIR sensors, as well as object tracking for five classes of objects. To the best of our knowledge, this is the first study of automated object detection and tracking using SWIR sensors.

### 2.2. Object Detection

Since deep convolutional neural networks were introduced in the computer vision field, many object detection approaches have been studied based on open datasets of visible-light wavelength images, such as COCO [13] or Pascal VOC [14]. Since YOLO [15] was introduced in 2016, the performance has been improved by its subsequent versions [12,16,17,18,19,20], and the latest version [20] was one of the state-of-the-art models in object detection. The YOLO models have provided lightweight models such as YOLOv5s and YOLOv7-tiny for devices of low computing power, and the lightweight models are being used in our research to deploy them on edge-computing devices for city surveillance.

Recently, transformer-based object detection models also showed state-of-the-art accuracies on the public datasets [21,22,23,24]. They used transformer encoder-decoder structures in the object detection problem to predict a set of boxes that were bipartitely matched to the ground truth object boxes. The recent transformer-based models showed higher accuracies than previous convolutional neural network-based object detection methods, but the inference speed was lower. In addition, there was much room for improvement in transformer-based models for use on edge-computing devices.

### 2.3. Multi-Object Tracking

Bewley et al. [25] proposed a multi-object tracking method called SORT, which used the Kalman filter [26] and the Hungarian algorithm [27] for target state management and box association, respectively. The authors used IoU distance scoring for the similarity measure. SORT was a simple and fast multi-object tracking algorithm, and many follow-up studies have been conducted based on SORT to overcome the weakness of instability at occlusion and the ID-switching problem. Zhang et al. [6] proposed ByteTrack, which associated object boxes with low confidence scores in the second round of the association. ByteTrack was based on the YOLOX [19] object detector, and the simple two-phase association method was effective.

Recently, similarity measures based on neural network features have been adopted in multi-object tracking. SmileTrack, proposed by Wang et al. [28], used a similarity learning module based on the Siamese network architecture, which extracted features from cropped areas of a person. The similarity score of the features was calculated using cosine similarity. Aharon et al. [29] proposed BoT-SORT using camera motion compensation and a re-identification method based on convolutional neural networks. The above approaches achieved state-of-the-art tracking accuracies while sacrificing speed for image cropping and feature extraction.

## 3. Study Approach

In this section, we present the statistics of our dataset of SWIR images, and the object detectors used in the research. In addition, the proposed tracking algorithm (MCDTrack) is explained.

### 3.1. Edge Computing

Deep neural networks require massive addition and multiplication operations that can be accelerated by GPU computing. As a result, training and inference of deep neural networks are typically performed on GPU-based servers. However, network bandwidth limitations or privacy issues often arise in city-surveillance scenarios, limiting the use of GPU server computing in such cases. Edge computing, which performs analysis close to the source of the data rather than on centralized servers, can be used in these situations.

Our city-surveillance scenario is one of the representative scenarios that require edge computing, as the complexity of resource management increases dramatically when streaming hundreds or thousands of high-resolution videos to central servers, and network bandwidth is often limited. In this study, we use the Raspberry Pi 4B, one of the most popular edge-computing devices, for object detection and tracking on SWIR images.

TFLite is a machine learning library developed by the Tensorflow community [30], for edge computing on low computing devices. Through the TFLite framework, a machine learning model is optimized to be run on edge-computing devices. The TFLite library provides the conversion of a Tensorflow model into a compressed flat buffer, and the quantization of 32-bit floating point (FP32) into 8-bit integers (INT8). We use the 8-bit quantized TFLite model, and we measure the running speed of the quantized object detection models on the Raspberry Pi 4B. The average running speeds of TFLite-optimized lightweight YOLO object detectors (YOLOv5s and YOLOv7-tiny) of 320 × 320 pixels input show 2.94 fps and 3.65 fps on the Raspberry Pi 4B. Data collection was performed based on the speed of these object detectors on Raspberry Pi 4B. The object detection and tracking experiments were also performed on the Raspberry Pi 4B.

### 3.2. Dataset

As SWIR sensors have rarely been investigated for automated object detection and tracking, we built a new SWIR dataset for city-surveillance scenarios. In the city center, a SWIR camera captured images of people, cars, trucks, buses, and motorcycles in a variety of environments, including streets, pavements, and crosswalks. The SWIR camera is installed in a fixed position with a top-down lateral angle, like standard surveillance cameras, and we vary the angle of view to capture different scenes.

Objects of the target classes are manually labeled by a labeling tool. Table 1 shows the statistics of our training and testing data. In total, 7309 images with 45,925 objects are used for training and 2689 images with 11,296 objects are used for testing. The resolution of SWIR images is 1296 by 1032 pixels.

In the test dataset, we also link the object boxes of successive time frames to create tracking labels. A total of 13 tracking scenarios are used to test the tracking algorithms in our research. A tracking scenario consists of consecutive image frames with a minimum duration of one minute. The average time gap between image frames is 327 ms (an average of 3.1 frames per second), which is relatively large compared to previously studied tracking datasets. To validate that our algorithm works well on edge-computing devices, which are slow in image acquisition, object detection inference, and object tracking, we sample the images at a low frame rate for experimental scenarios. This low frame rate is appropriate for the speed of the object detection algorithm on the Raspberry Pi 4B. The number of image frames and the number of objects in each scenario are shown in Table 2.

### 3.3. Detecting Objects in SWIR Images

The main purpose of this research is to detect and track objects in SWIR images of city-surveillance scenarios. In addition, we hope that the surveillance algorithm requires low computational resources so that it can be deployed on edge-computing devices. We use lightweight models of recent object detectors, including YOLOv5s and YOLOv7-tiny, and we also use TFLite to accelerate the computation of neural networks.

Table 3 shows the number of parameters and multiply–add operations in GFlops. Compared to the YOLOv7 base model with 37.22 M parameters and 52.43 GFlops, the YOLOv7-tiny model has only 16.2% of parameters and 12.6% of GFlops. We train the object detection models on the previously mentioned training data of SWIR images for 300 epochs from scratch. These lightweight models are further optimized by quantization into 8-bit integer weights and operations.

### 3.4. Tracking Object in SWIR Images

Algorithm 1 shows the pseudo-code of our MCDTrack (Multi-class Distance-based Tracking). MCDTrack is given a tracking scenario consisting of consecutive image frames, an object detector, distance thresholds for each class, and two confidence thresholds $\tau_{high}$ and $\tau_{low}$ representing high and low confidence levels used in the algorithm. The object detector receives an image frame and returns object box information, which includes box coordinates, a class, and a confidence score.

For each frame in a tracking scenario, MCDTrack retrieves each class to associate the bounding boxes of the class. Similar to previous multi-object tracking approaches [6,28,29], the detection boxes of a given class are divided into high-confidence boxes and low-confidence boxes along the two confidence thresholds $\tau_{high}$ and $\tau_{low}$. The high-confidence boxes are object boxes with confidence thresholds higher than $\tau_{high}$, and the low-confidence boxes are those with confidence thresholds between $\tau_{high}$ and $\tau_{low}$. The first association uses the high-confidence boxes, and the second association uses the low-confidence boxes.
**Algorithm 1:** Pseudo-code of MCDTrack.
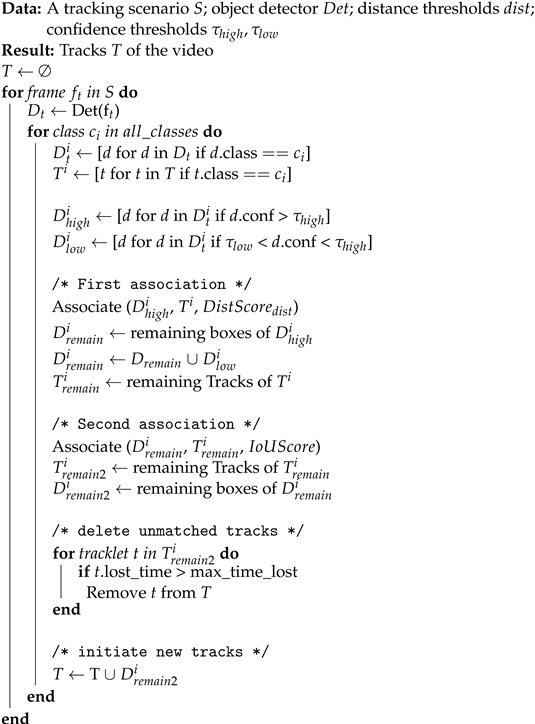


In MCDTrack, a distance-based similarity score is used for the first association as shown in Algorithm 1. The distance-based similarity score measures the distance of the centers between object boxes (*D*) and tracks (*T*). The center of each object box and track is represented by ((xmin+xmax)/2,(ymin+ymax)/2). We then calculate the Euclidean distances of the centers, and the result is a matrix of |D|×|T|, where each cell in (i,j) represents the distance between *i*’th object box and *j*’th track. We then divide the distance matrix by the distance threshold of each class, and the pairs that exceed the distance thresholds receive the highest distance cost and are not assigned to each other in the association phase. The distance threshold for each class is determined empirically, and the values in pixels are as follows—Person: 350, Car: 100, Motorcycle: 250, Bus: 100, Truck: 100. We found that the person and motorcycle classes should have longer matching distance thresholds because they are more likely to be occluded and move faster than objects of other classes, as shown in the example of Figure 1. The distance-based score assumes that the boxes of the same object are closer together in successive time frames than boxes of different objects.

In conjunction with the distance-based score, the IoU-based score is utilized as the second association. The IoU score is used in many previous multi-object tracking approaches [6,25], and it measures the intersection over the union of detected boxes and previous tracks. It assumes that boxes of the same object in consecutive time frames overlap more than boxes of different objects. The IoU score is simple and effective in the multi-object tracking problem, but there could be no overlap if a target object is moving fast. The distance and IoU methods are simple to implement and do not require additional feature extraction and region-of-interest cropping from feature maps of convolutional neural networks, which could reduce tracking speed. Using the above similarity scores, a matrix is created showing how likely it is that object boxes and tracks are from the same object. The Hungarian algorithm is then used to find an optimal match in the matrix.

Compared to previous multi-object tracking approaches, the characteristics of our MCDTrack algorithm can be summarized as follows: (1) The Kalman filter is not used to predict the location and size of the boxes. Previous tracking approaches used the Kalman filter with a single predefined setting of mean and covariance matrices, which is not appropriate in our scenario of multi-class multi-object tracking scenario. The sizes, speeds, and width/height ratios vary for each class, and a single Kalman filter was unable to accurately predict the tracking box information. (2) Instead of visual feature-based re-identification scores, we use distance and IoU scores for the similarity measure. Visual feature-based methods require additional neural network computation. In addition, cropping the region of interest and comparing the high-dimensional features can reduce the latency of the algorithm, and it is not suitable for edge computing on regional surveillance cameras. (3) We adopt a multi-class tracking approach, which has not been thoroughly investigated in previous multi-object tracking approaches. In our MCDTrack, the pairs of track and bounding boxes of the same classes are matched. In addition, we investigate the best setting of distance thresholds for each class, which is an effective way to improve tracking performance.

The key factors in our MCDTrack are that it uses a two-phase association method as in the previous work [6], and it uses distance-based tracking for the first association, which is fast and effective, especially in an edge-computing environment of low frame rate and low computing power. The Euclidean distance computation requires only a small number of multiplications, and the distance computation of multiple track information and multiple object boxes could be further improved by recent libraries, such as SciPy [31]. The distance-based similarity measure allows the association of objects of high speed that do not overlap in successive frames, and our class-specific distance threshold approach is also effective to consider objects of different speeds.

Figure 2 shows an example of tracking objects in images of consecutive time frames. Most of the objects have a high overlap between the boxes in the *t* and *t* + 1 time frames, but the red (a person) and orange (a motorcycle) boxes have no overlap between them. Because these objects are faster and smaller than other objects, it is difficult to track these objects using only the IoU measure. Figure 3 also gives an example of how the width-to-height ratios of identical objects can vary significantly. The bus appears as a vertically long bounding box, but after 14 time steps, it becomes almost square in shape. Unlike the pedestrian tracking dataset, the urban surveillance scenarios in our research contain such significant changes in width-to-height ratio that it is difficult to apply a fixed Kalman filter setting.

Similar to previous work on object tracking [6,28], we use representative multi-object tracking metrics [32,33], such as Multi-Object Tracking Accuracy (MOTA), the number of false positives, the number of false negatives, the number of switches, identification precision (IDP), identification recall (IDR), identification f1 score (IDF1), etc. Among these, MOTA and IDF1 are used as representative metrics. MOTA is defined by subtracting the proportion of false negatives, the proportion of false positives, and the proportion of mismatches from one [32]. IDF1 is defined by the harmonic mean of identification precision and identification recall [33]. Although MOTA focuses more on object detection accuracy than the IDF1 metric, IDF1 places more weight on how the tracking association matches with the ground truth than MOTA. Since MOTA and IDF1 are defined in the single-class experimental setting, we also measure the mean of MOTA (mMOTA) and the mean of IDF1 (mIDF1) by averaging the MOTA and IDF1 results for each class.

## 4. Study Results

In this section, the object detection and tracking accuracies on the SWIR image dataset are presented. Ablation studies on parameters of our MCDTrack are also conducted on object detection models and confidence thresholds.

### 4.1. Detecting Objects in SWIR Images

Table 4 shows mAP results on the test dataset. The object detection models are trained in a desktop-computing environment equipped with a high-performance GPU, and the model is exported to an int8-quantized TFLite model. Then the experiments are conducted on Raspberry Pi 4B with input images of resized to 320 × 320 pixels. The models show 80.1% to 86.3% of mAP[0.5] results, which are high enough to be used in city-surveillance scenarios. More specifically, YOLOv7-tiny model shows higher average precision (AP) values in every class. YOLOv7 [20] adopted extended efficient layer aggregation networks, concatenation-based scaling, and planned re-parameterized convolution, making it more efficient than its predecessor, YOLOv5 [18].

### 4.2. Tracking Objects in SWIR Images

In this subsection, multi-object tracking accuracy of MCDTrack on the SWIR object tracking dataset is presented. We attempted to compare our models with previous work on multi-object tracking using the SWIR object tracking dataset. However, the comparison was inappropriate as previous work has mainly focused on single-class multi-object tracking on visible-wavelength images. Even when using the detection results from our object detectors, previous Kalman filter-based models, such as ByteTrack [6], were unable to track objects due to differences in moving speeds and width/height ratios between different classes—the MOTA results are negative because of too many false positives and false negatives. Fine-tuning the parameters for each class would improve the performance of such models, but is beyond the scope of this research. Therefore, we compare our model with the ByteTrack [6] without Kalman filter while the same object detection models for SWIR images are used. In addition, we also perform ablation studies on the association methods and parameters of the association methods.

#### 4.2.1. An Ablation Study on Association Methods and Object Detection Models

Table 5 shows the tracking accuracy results of different experimental settings. We compare ByteTrack [6] without Kalman filter, trackers of different first and second association methods, and our MCDTrack. For example, in Table 5, *IoU/IoU* represents that the IoU score is utilized in both the first and second association. Our MCDTrack utilizes the distance score with distance thresholds by class for the first association, and the IoU score is used for the second association.

In Table 5, the following basic experimental setting is used—$\tau_{high}=0.5$, $\tau_{low}=0.1$, dist=500, dist2=100 (dist2 is used in only in the tracker with *distance/distance*). Our MCDTrack uses $\tau_{high}=0.7$, $\tau_{low}=0.1$, and the distance thresholds for each class that are explained in Section 3. In Table 5, the red font color presents the best results. The meaning of column names is as follows—False Positives (FP), False Negatives (FN), Identification Switching (IDs), Multi-Object Tracking Accuracy (MOTA), Identification Precision (IDP), Identification F1 score (IDF1), mean of MOTA (mMOTA), and mean of IDF1 (mIDF1). Lower values of negative metrics (FP, FN, and IDs) and higher values of other metrics represent better tracking results.

First, we found that YOLOv7-tiny is superior to YOLOv5s object detector. MOTA, IDF1, mMOTA, and mIDF1 are higher in MCDTrack with YOLOv7-tiny than in with YOLOv5s. As shown in Table 4, the mAP is superior in YOLOv7, and the tracking accuracy also follows a similar trend because tracking is conducted by associating detected object boxes. We also find that our MCDTrack outperforms ByteTrack by a large margin in all types of metrics in both object detectors. ByteTrack shows too many false negatives which result in low MOTA and mMOTA. Our MCDTrack also shows 4.4% higher IDP and 14.8% higher in IDR, which results in 10.5% higher IDF1 when YOLOv7-tiny is used.

When comparing the association methods on the YOLOv7-tiny object detector, MOTA values were higher in the order of IoU/IoU, IoU/Distance, Distance/Distance, and Distance/IoU methods in our SWIR dataset. When the first association was changed from IoU to Distance, there was an improvement of 1.7% to 2.4% in MOTA. Although the values of FP and FN did not show a significant difference, the results show that ID switches were significantly reduced when the first association was changed from IoU to Distance. This result suggests that matching the high-confidence boxes with the IoU similarity measure can produce more ID switching, which represents the failure of matching object boxes of the same object in successive time frames. Considering these experimental results, our MCDTrack uses Distance/IoU similarity measures, and we further improve the algorithm by adopting distance thresholds for each class.

#### 4.2.2. An Ablation Study on Confidence Thresholds

In Figure 4, we examine MOTA, IDF1, mMOTA, and mIDF1 results from our MCDTrack while varying two confidence thresholds ($\tau_{high}$ and $\tau_{low}$). The *x*-axis represents the value of $\tau_{high}$ (values range from 0.2 to 0.9) and the *y*-axis represents the value of $\tau_{low}$ (values range from 0.1 to 0.5). The higher values are represented by darker colors. In MOTA, YOLOv5s shows the best result of 76.0 with $\tau_{high}$ of 0.8 and $\tau_{low}$ of 0.1, and YOLOv7-tiny shows the best result of 78.0 with $\tau_{high}$ of 0.7 and $\tau_{low}$ of 0.4. In IDF1, YOLOv5s shows the best result of 79.1 with $\tau_{high}$ of 0.8 and $\tau_{low}$ of 0.1, and YOLOv7-tiny shows the best result of 80.2 with $\tau_{high}$ of 0.7 and $\tau_{low}$ of 0.1. mMOTA and mIDF1 show similar trends with MOTA and IDF1. The results indicate that high values (0.7∼0.8) of $\tau_{high}$ are adequate to match high-confidence boxes, and it is valuable to inspect low-confidence boxes, which have confidence values from 0.1 to 0.7, in the second association.

#### 4.2.3. Tracking Speed

We compare the tracking speed of our MCDTrack and ByteTrack [6]. Both are based on fast and simple scoring algorithms. We compare the average tracking times of ten iterative runs on our tracking scenarios of SWIR images. This experiment is performed on the Raspberry Pi 4B. On average, ByteTrack takes 4.22 ms, while our MCDTrack takes 2.53 ms, which is 40.0% less. The tracking time is much less than the average object detection time because the computation of IoU and distance takes less time than the computation of convolutional neural networks, but the 40% reduction in tracking time will be significant for edge devices that do not have enough computing power.

#### 4.2.4. Qualitative Analysis

Figure 5 and Figure 6 show qualitative examples of our MCDTrack and ByteTrack. In Figure 5, the results of our MCDTrack (left side) keep track of two people, while ByteTrack fails to track them because they are occluded by a tree in the middle, which cannot be tracked by the IoU method only. In Figure 6, our MCDTrack shows a better performance than ByteTrack in complicated tracking situations. In the upper figure, the motorcycle is not tracked by ByteTrack, but our MCDTrack keeps the object on track. In the figure below, ByteTrack misses the tracks of some of the people, while our MCDTrack gives a much better result for the tracking of the people.

## 5. Conclusions

In this paper, a simple, fast, and effective multi-object tracking method is proposed, called MCDTrack. Previous multi-object tracking studies have mainly focused on tracking people or animals in high-frame-rate visible-wavelength images, but our study focuses on object detection and tracking in low-frame-rate SWIR images for use in edge-computing environments. Our experiments on the SWIR image dataset from city surveillance show that the proposed MCDTrack is superior to previous multi-object trackers. The lightweight object detectors are studied, and the 8-bit integer quantized of them are utilized in the edge-computing device for our experiments.

Our MCDTrack shows 77.5% in MOTA and 80.2% in IDF1, although the frame rate is low compared to the previous tracking dataset and the object detection accuracy is limited because the detection results are derived from the edge-computing environment, and this indicates that our MCDTrack with the lightweight object detector can be easily applied to real-world applications. The proposed method on edge-computing devices will be very useful especially when the network environment is not sufficient to send all video streams from the regional cameras to the servers, or when sending video streams is not available due to security and privacy reasons. Furthermore, the results of this study can be used in situations where city surveillance is required in low light or foggy conditions, where SWIR sensors outperform visible-wavelength sensors. In the future, different types of edge-computing environments, especially those incorporating NPU or GPU to accelerate neural network processing, will be investigated for use in automated object detection and tracking. A system design that utilizes multiple cameras and edge computers working together to track objects is another way to extend our research.

## Figures and Tables

**Figure 1 sensors-23-06373-f001:**
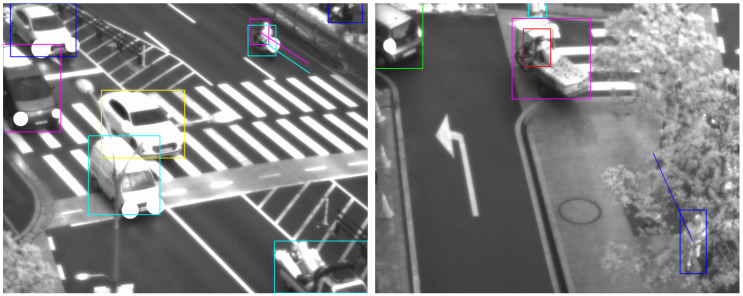
An example of a long matching distance between an object from two different frames. The lines indicate the distances from previous tracking information. (**left**) A motorcycle (cyan color) and a person (purple color) move faster than other objects. (**right**) A person (blue color) is occluded by a tree.

**Figure 2 sensors-23-06373-f002:**
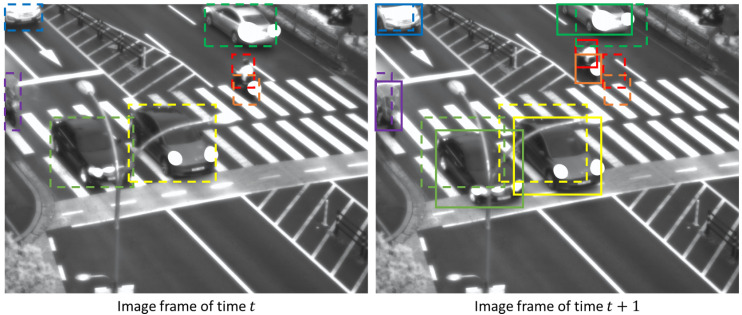
An example of tracking objects in images of consecutive time frames. Bounding boxes of the image frame at time *t* are presented in dashed lines, and those at time *t* are presented in solid lines. The boxes of a person (red color) and a motorcycle (orange color) do not overlap, while other boxes overlap in consecutive time frames. Best viewed in color.

**Figure 3 sensors-23-06373-f003:**
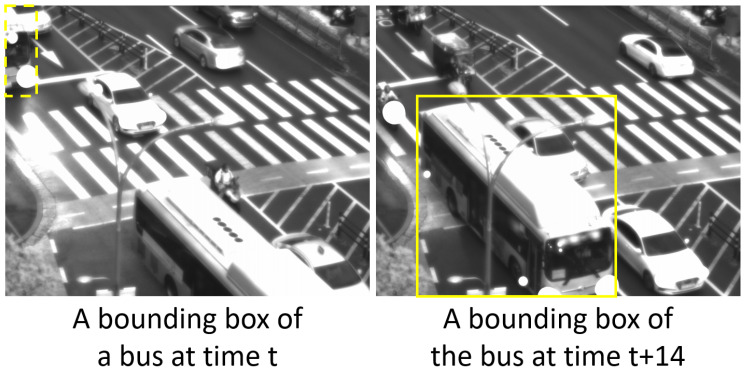
An example of how the width-to-height ratio of the same object can change significantly. The bounding box of a bus at time *t* is shown in the dashed yellow box (the top left corner of the left image), and the bounding box of the bus at time *t* + 14 is shown in the solid yellow box (center of the right image).

**Figure 4 sensors-23-06373-f004:**
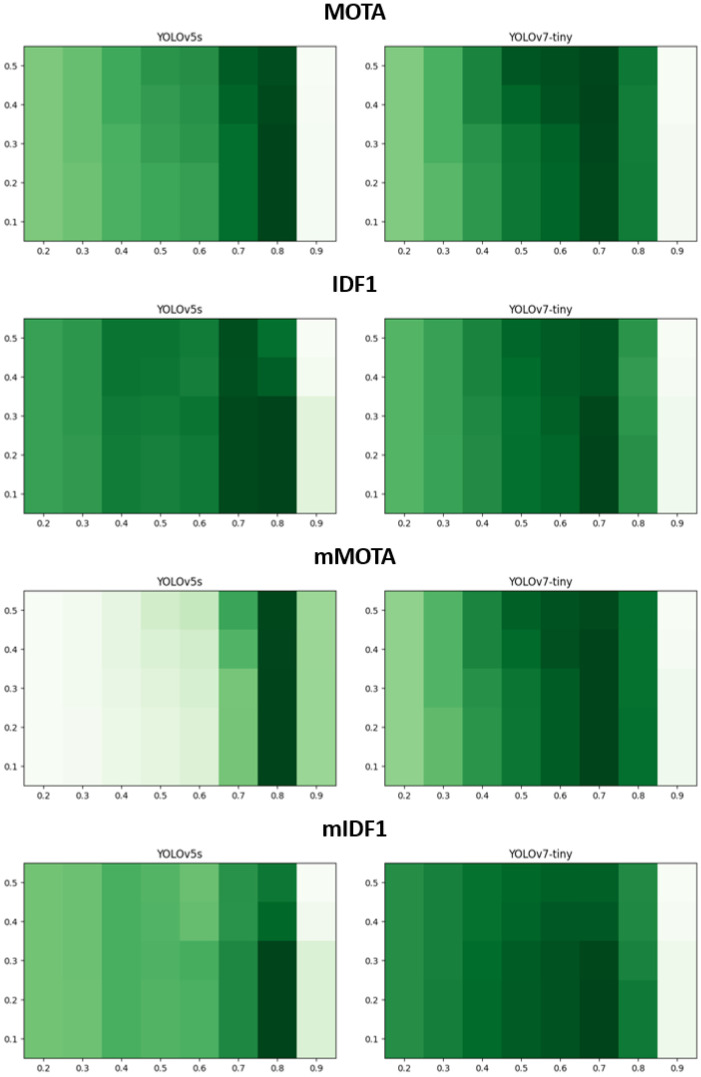
MOTA, IDF1, mMOTA, and mIDF1 results while varying the confidence thresholds, $\tau_{high}$ and $\tau_{low}$. The *x*-axis represents $\tau_{high}$ values, and the *y*-axis represents $\tau_{low}$ values. A darker color represents a higher value.

**Figure 5 sensors-23-06373-f005:**
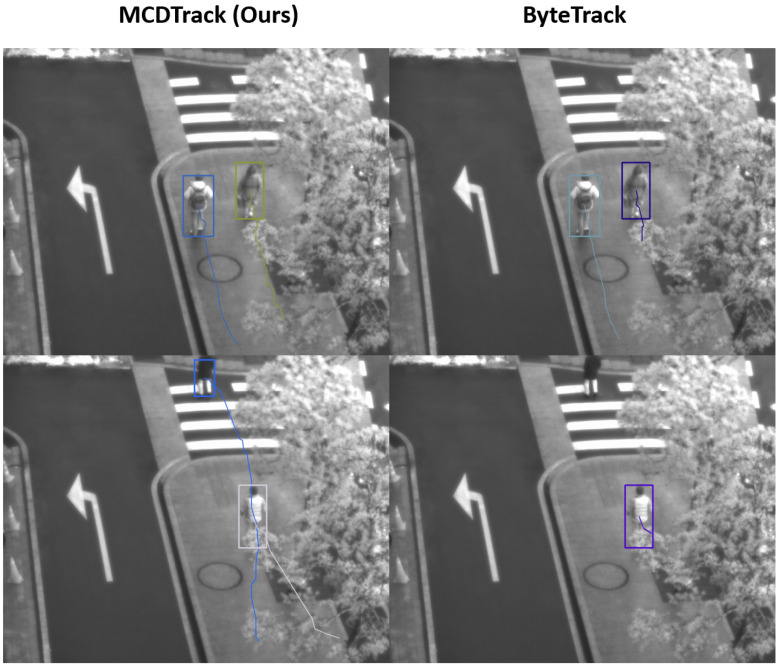
Qualitative examples of the proposed MCDTrack and ByteTrack. MCDTrack is more robust to occlusion and tracking of long-range motion. (top) MCDTrack was able to correctly track a person (dark yellow color), while ByteTrack failed to connect the route before occlusion (purple color). (bottom) MCDTrack is robust to occlusion and tracks two people correctly (blue and white colors), but ByteTrack failed to track them. (purple color).

**Figure 6 sensors-23-06373-f006:**
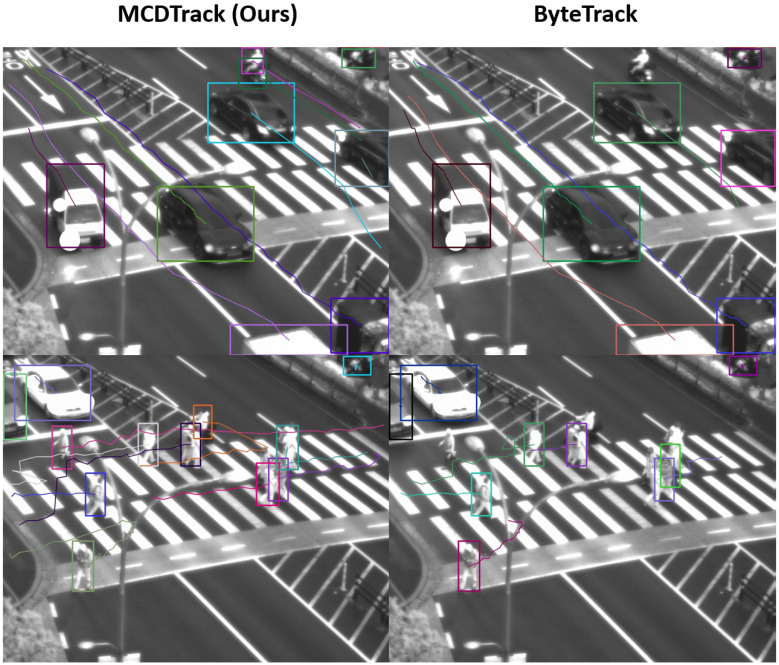
Qualitative examples of the proposed MCDTrack and ByteTrack. MCDTrack shows better tracking performance in complicated tracking situations. (top) MCDTrack was able to track the motorcycle and the person (purple and dark green colors) while ByteTrack failed. (bottom) ByteTrack failed to track three people who are correctly tracked by MCDTrack (orange, pink, and peacock colors).

**Table 1 sensors-23-06373-t001:** The number of objects in training and testing datasets.

	Training	Testing
images	7309	2689
objects		
person	13,582	4037
car	18,617	4548
motorcycle	7383	1633
bus	1313	260
truck	5030	818
All	45,925	11,296

**Table 2 sensors-23-06373-t002:** The number of frames and the number of objects in each tracking scenario.

	# of Frames	# of Objects
Scenario 1	168	200
Scenario 2	173	213
Scenario 3	183	307
Scenario 4	193	450
Scenario 5	209	422
Scenario 6	207	542
Scenario 7	206	370
Scenario 8	298	815
Scenario 9	211	1806
Scenario 10	204	1247
Scenario 11	210	1557
Scenario 12	211	1305
Scenario 13	216	2062
All	2689	11,296

**Table 3 sensors-23-06373-t003:** The number of parameters and GFlops of three studied object detection models.

	Params (M)	GFlops
YOLOv7	37.22	52.43
YOLOv5s	6.62	8.43
YOLOv7-tiny	6.03	6.59

**Table 4 sensors-23-06373-t004:** AP and mAP Results of Object Detectors.

	YOLOv5s	YOLOv7-Tiny
Person	72.9	**75.4**
Car	94.8	**97.4**
Motorcycle	88.4	**88.8**
Bus	61.2	**86.2**
Truck	83.0	**83.6**
mAP[0.5]	80.1	**86.3**

**Table 5 sensors-23-06373-t005:** Tracking results of different tracker and object detector settings. Red colors indicate that the best result in each column.

Tracker	Detector	FP	FN	IDs	MOTA	IDP	IDR	IDF1	mMOTA	mIDF1
ByteTrack (w/o KF)	yolov5s	1350	3404	218	56.0	75.1	61.5	67.7	33.8	61.2
ByteTrack (w/o KF)	yolov7-tiny	865	3336	285	60.3	79.3	62.0	69.7	38.6	60.5
IoU/IoU	yolov5s	1614	1610	417	67.8	75.6	75.7	75.7	51.1	71.7
IoU/IoU	yolov7-tiny	1510	1270	516	70.8	74.6	76.2	75.4	72.2	78.7
IoU/Distance	yolov5s	1632	1603	289	68.8	75.7	76.0	75.9	52.0	72.3
IoU/Distance	yolov7-tiny	1634	1251	348	71.4	74.8	77.4	76.1	72.9	79.6
Distance/Distance	yolov5s	1601	1614	227	69.5	74.3	74.3	74.3	52.5	71.5
Distance/Distance	yolov7-tiny	1480	1292	269	73.1	74.5	75.8	75.2	73.9	79.0
Distance/IoU	yolov5s	1610	1611	226	69.5	74.2	74.2	74.2	52.4	71.5
Distance/IoU	yolov7-tiny	1493	1277	260	73.2	74.8	76.3	75.6	73.9	79.3
Ours (MCDTrack)	yolov5s	243	2319	152	76.0	87.9	71.8	79.1	71.8	77.9
Ours (MCDTrack)	yolov7-tiny	694	1634	216	77.5	83.7	76.8	80.2	77.2	81.6

## Data Availability

Not applicable.

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
