# Peer review of "Multi-Object Tracking on SWIR Images for City Surveillance in an Edge-Computing Environment"

_sensors, 2023, doi:10.3390/s23146373_

Round 1

Reviewer 1 Report

Deep neural networks have greatly enhanced object detection and tracking systems, particularly in the context of automatic recognition systems like military and urban surveillance. While previous studies primarily focused on visible image data sets, recent research has highlighted the potential of Short-Wave Infrared (SWIR) sensors for target detection and tracking. SWIR sensors offer advantages such as the ability to see through fog and thin layers due to their sensitivity to longer light wavelengths. However, SWIR images have not been extensively explored for object detection and tracking algorithms. The proposed paper suggests MCDTrack, a multi-class distance-based tracking approach, specifically designed for city surveillance using SWIR images and edge computing devices like Raspberry Pi 4B. The proposed method achieves promising results and can be applied in real city surveillance scenarios.

Related work section deals with the use of  SWIR sensors for automated object detection and tracking, as well as research trends in object detection and multi-object tracking. Various studies have explored the application of SWIR sensors in different scenarios, including mud area detection, facial recognition, and moving object detection. Object detection has seen significant advancements with the introduction of deep convolutional neural networks, such as YOLO models, which provide lightweight options for low-power devices like edge computing devices. Transformer-based object detection models have also shown promise but have slower inference speeds and room for improvement in edge computing applications. Multi-object tracking methods like SORT and ByteTrack have been proposed, and recent approaches have incorporated neural network features for similarity measurement.

The main conclusion of the paper is that the proposed multi-object tracking method, MCDTrack, is simple, fast, and effective for object detection and tracking in low frame rate SWIR images in edge computing environments. The experiments conducted on a SWIR image dataset from city surveillance demonstrate that MCDTrack outperforms previous multi-object trackers. Despite the limitations of low frame rate and limited object detection accuracy in the edge computing environment, MCDTrack achieves promising results, with 77.5% in MOTA and 80.2% in IDF1. The study suggests that MCDTrack, along with lightweight object detectors, can be easily applied to real-world applications. The proposed method is particularly valuable in scenarios where network constraints or security and privacy concerns prevent the transmission of video streams, and in conditions where SWIR sensors offer advantages over visible wavelength sensors, such as low light or foggy environments.

The cited literature is relevant to the field and mostly of recent date. There is a list of used abbreviations at the end of the manuscript.

Some minor issues that have to be adressed:

1.       Abstract is too short, it should be expanded to 200 word.

2.       Couple of more keywords should be added.

3.       Discription below the figures should be more general. The authors should try not to use phrases like Ours

Reviewer 2 Report

The manuscript proposes a fast and effective multi-object tracking method called MCDTrack for SWIR images in city surveillance scenarios. The paper addresses the lack of research on automated object detection and tracking systems using SWIR images. The proposed MCDTrack algorithm is designed to be efficient for deployment on low-power and low-computation edge computing devices. The experimental results demonstrate the superiority of MCDTrack over previous multi-object tracking methods and its high tracking accuracy. The manuscript is well-structured and provides valuable insights into the development of robust city surveillance solutions in edge computing environments.

The introduction effectively highlights the limitations of existing object detection and tracking systems when applied to SWIR images. However, it would be beneficial to provide more specific details on the advantages of SWIR sensors, such as their ability to penetrate atmospheric conditions, robustness in low light, and their potential impact on city surveillance applications. This would help the reader better understand the motivation behind exploring SWIR images for object detection and tracking.

In the methodology section, provide a more detailed explanation of the MCDTrack algorithm. Describe the key steps, components, and techniques involved in the proposed method. Clarify how MCDTrack utilizes distance-based tracking and provide insights into how it achieves its fast and effective multi-object tracking performance.

In the results section, include a comparison with other relevant state-of-the-art multi-object tracking methods, especially those specifically designed for SWIR images or edge computing environments. This will strengthen the evaluation of MCDTrack and provide a more comprehensive understanding of its performance.

The language and style of the manuscript are generally clear and concise, effectively conveying the key ideas and findings.

Reviewer 3 Report

The article describes the multi-object tracking method on SWIR images for city surveillance in an edge computing environment and presents its limitations. In the work, the authors correctly presented the issues related to the detection and tracking of the objects and the methods that are used for this purpose. This presentation was supported by appropriately selected literature. The methods used so far are presented and the solution proposed by the authors is described in detail. The results of the work were verified by the experimental research. On the basis of correctly presented analysis of the research results, complete conclusions were drawn. The reviewed paper is well organized with an introduction, the theoretical part, the presentation of research results and the conclusions. After the theoretical part, the study presents the results of the research and their analysis.

Recommendations for improving the manuscript:

1. At the beginning of the article, it should be specified what is its purpose and scope of work (what is the work of the authors).

2. Has the presented method been tested in typical practical applications or only in research works?

3. What are the research plans for the future in this area?
